

# Relationship between the sub-micron fraction (SMF) and fine mode fraction (FMF) in the context of AERONET retrievals

Norman T. O'Neill[1], Keyvan Ranjbar[2], Liviu Ivănescu[1], Thomas F. Eck[3], Jeffrey S. Reid[4], David M. Giles[3], Daniel Pérez-Ramírez[5], Jai Prakash Chaubey[6]

[1]Centre d'applications et de recherches en télédétection (CARTEL), Université de Sherbrooke, Sherbrooke, J1K 2R1, Canada
[2]Flight Research Laboratory, National Research Council, Ottawa, ON, K1V 1J8, Canada
[3]NASA Goddard Space Flight Center, Greenbelt, MD 20771, USA
[4]US Naval Research Laboratory, Monterey, CA 93943, USA
[5]Applied Physics Department, University of Granada, Granada 18071, Spain
[6]Department of Physics and Atmospheric Sciences, Dalhousie University, Halifax, NS, B3H 4R2, Canada

*Correspondence to*: Norman T O'Neill (Norman.T.ONeill@usherbrooke.ca)

**Abstract.** The sub-micron (SM) aerosol optical depth (AOD) is an optical separation based on the fraction of particles below a specified cut off radius of the particle size distribution (PSD) at a given particle radius. It is fundamentally different from spectrally separated FM (fine mode) AOD. We present a simple (AOD-normalized) SM fraction versus FM fraction

(SMF vs FMF) linear equation that explains the well-recognized empirical result of SMF generally being greater than the FMF. The AERONET inversion (AERinv) products (combined inputs of spectral AOD and sky radiance) and the Spectral Deconvolution Algorithm (SDA) products (input of AOD spectra) enable, respectively, an empirical SMF vs FMF comparison at similar (columnar) remote sensing scales across a variety of aerosol types.

SMF (AERinv derived) vs FMF (SDA derived) behavior is primarily dependent on the relative truncated portion ($\varepsilon_c$)

of the coarse mode (CM) AOD associated with the cutoff portion of the CM PSD and, to a second order, the cutoff FM PSD and FM AOD ($\varepsilon_f$). The SMF vs FMF equation largely explains the SMF vs FMF behavior of the AERinv vs SDA products as a function of PSD cutoff radius ("inflection point") across an ensemble of AERONET sites and aerosol types (urban industrial, biomass burning, dust, maritime and a mixed class of Arctic aerosols). The overarching dynamic was that the linear SMF vs FMF relation pivots clockwise about the approximate (SMF, FMF) singularity of (1, 1) in a "linearly inverse" fashion (slope

and intercept of approximately $1 - \varepsilon_c$ and $\varepsilon_c$) with increasing cutoff radius. SMF vs FMF slopes and intercepts derived from AERinv and SDA retrievals confirmed the general domination of $\varepsilon_c$ over $\varepsilon_f$ in controlling that dynamic. A more general conclusion is the apparent confirmation that the optical impact of truncating modal (whole) PSD features can be detected by a SMF vs FMF analysis.

## 1 Introduction

Anderson et al, (2005) noted the "decades old observation that aerosol mass generally consists of two modes: (1) a mechanically produced coarse mode [CM] and (2) a fine mode [FM] produced by combustion and/or gas to particle





conversion". Typical CM examples include wind-eroded desert dust and sea-salt while FM aerosols tend to be dominated by biomass-burning smoke and anthropogenic and biogenic (ABF) fine particles (the latter classification being proposed, for example, by Lynch et al., 2016). Given this speciated physical reality (and notwithstanding the known chemical and

microphysical internally mixed changes that occur in aerosol properties as they are transported through the atmosphere) there is a certain level of justification in treating different species of aerosols as independent FM and CM particle size distributions (PSDs). The degree of rigour in this modal paradigm is, in fact, the subject of model analyses that compare prescribed bulk aerosol PSDs with sectional (binned) PSDs (see, for example, Mann et al., 2012 for the case of size-referenced PSDs and Kodros and Pierce, 2017 for the case of mass-referenced PSDs).

AERONET inversions (AERinv), derived from input spectral AODs and solar almucantar radiances (Dubovik and King, 2000), provide a comprehensive suite of microphysical and optical products (at ~ hourly temporal resolutions). The general tendency towards (particle-volume) PSD bimodality is notably evidenced in retrieved AERinv PSDs (see, for example, Figure 1 of Dubovik et al., 2002, Figure 11a of Eck et al., 2009 and Figure 3 of AboEl-Fetouh et al., 2020[1]). A bi-modal optical representation of that bimodality (without the requirement of having to specify the PSD shape of each mode) can largely

determine the measurable (low spectral order) optical behavior of remote sensing data (O'Neill et al, 2001 and references cited therein). O'Neill et al. (2003) employed spectral AODs (sampling intervals ~ 3 minutes) as an input to their spectral deconvolution algorithm (SDA) to separate aerosols into (a) extensive (quantity dependent) FM and CM AOD components ($\tau_f$ and $\tau_c$) of the total AOD ($\tau_a$), (b) intensive (quantity independent) spectral derivatives of $\tau_f$ ($\alpha_f, \alpha_f', \alpha_f''$, etc.) and $\tau_c$ ($\alpha_c, \alpha_c', \alpha_c''$, etc.) as components of the total AOD spectral derivatives ($\alpha, \alpha', \alpha''$, etc.) as well as (c) semi-intensive FM and

CM fractions (FMF and CMF represented respectively by $\eta = \tau_f/\tau_a$ and $1 - \eta = \tau_c/\tau_a$). If aerosols can be viewed as independent coarse and fine modal features then this purely spectral technique (which is generically assigned the FMF acronym) acts to separate those modal features in an optical sense.

The SMF (sub-micron fraction) is a microphysically determined alternative to the optically determined FMF: the division into fine and coarse components is effected by an explicit separation of the PSD at a cutoff radius ($r_0$) that typically

ranges from ~ 0.4 to 1.1 μm across different types of aerosol instruments[2][3] (a similar radius range defines the AERinv case). This approach represents a moderate but significant difference relative to the optically based FMF separation. Anderson et al. (2005) describe the SMF as " …an operational definition … to distinguish it from the theoretical concept of fine mode fraction, FMF." Because the AERinv approach incorporates a cutoff radius division of the retrieved PSD into sub- and super-micron parameters (Dubovik and King, 2000) it is actually an SMF approach. Comparisons with the AERONET SDA product provide

---

[1] The examples in these papers can include the apparent presence of more than two modes (notably what often appear to be two CM sub-modes). Such cases are discussed below.
[2] The expression "sub-micron" is generally "defocussed" (relative to a literally exact value of 1.0 μm) to encompass this approximate range of $r_0$ values.
[3] It is worth noting that the in situ community almost universally refers to diameter rather than radius.





a unique opportunity to empirically analyze the SMF vs FMF approaches at similar remote sensing (columnar) scales and across a global variety of aerosol types.

The literature on the direct comparison of the SMF vs FMF retrievals is sparse and oftentimes at the margins of significance. The significance problem relates to the fact that one may be forced to extract relatively subtle microphysical and optical changes in the face of SMF and FMF variations that are often limited in range (lack of extensive-parameter aerosol

variation and / or types for example). The sparsity of SMF vs FMF investigations relates to limitations such as the rarity of experiments designed for such a comparison or the sparsity of data sets whose modes can be readily separated (by source information and/or chemical identification for example).

In the most controlled experiments, volume-sampled SMF estimates are compared with FMF retrievals derived from applying the SDA to volume-sampled, 3-channel CRD (Cavity Ring Down) or 3-channel nephelometer "spectra" (Atkinson

et al., 2010, Kaku et al., 2014 (Ka) and Atkinson et al., 2018). In general we find, as expected, the SMF to be >~ FMF in Atkinson et al., 2010 and Ka: this is notably true at the near-IR (700 nm) wavelength in Ka (ACE-Asia data of their Figure 2)[4]. The cases where this is not true (the T1 "ext_sum" results of Atkinson et al., 2018) are likely attributable to the spectral sensitivity of the FMF: this is most severe in the case of just 3 channels (for example, Ka's VOCALS 5% calibration correction of one 450 nm nephelometer channel transformed a case of SMF being substantially < FMF to a corrected case of SMF ~

FMF).

Lesser constrained experiments involve multi-altitude, volume-sampled SMF estimates (from nephelometer scattering coefficient & absorption coefficient devices) compared with layer and column SDA-derived FMF estimates from multi-altitude AOD and ΔAOD spectra acquired with an airborne sunphotometer (for example, Gassó & O'Neill, 2006 and Shinozuka et al., 2011; respectively represented by the acronyms GO and SHN). In these cases, the lesser degree of experimental control

(associated with multi-altitude flights) was somewhat offset by a more generous number of sunphotometer spectral bands (4 bands from the UV to the NIR in the GO case and 5 bands near the 5 bands employed in the AERONET SDA in the SHN case). The GO results were fairly coherent with SMF / FMF expectations (especially when the layered estimates were added to obtain columnar estimates) and quite marginal for SHN with a large spread of near-unity points about the SMF = FMF line (a situation with practically no significant SMF or FMF range and hence little relevant testing of their relationship). Analyses

of the FMF vs SMF relationship comparing satellite estimates of FMF[5] with, for example, airborne estimates of SMF (Anderson et al., 2005) or coastal/island AERinv estimates of SMF (Kleidman et al., 2005) have proven difficult given the spectrally sensitive nature of the few bands employed in satellite-derived estimates of FMF and all the possible sources of

---

[4] for which the fixed cutoff radius for the optically smaller particles (and thus more optically active in the sense of more strongly attenuating) results in a greater relative contribution to the FM AOD (by "optically smaller" we mean the ratio of particle size to wavelength has decreased because the wavelength has increased). The optically smaller particles are more optically active because they are ascending the right side of the anomalous diffraction peak (see, for example, O'Neill et al., 2005).

[5] retrievals that generally employ prescribed and speciated bulk (modal) FM and CM PSDs (of constant shape and position as a function of radius) as a basis for fitting spectral AODs at a few wavelengths.





incoherencies between the satellite retrievals and the ground- or airborne estimates of SMF (differences in spatio-temporal sampling volumes for example).

A simple (approximate) relationship between the SMF and FMF will be presented below. We seek to demonstrate that the AERONET SMF and FMF versions are largely governed by that relationship and that fitting parameters extracted from their empirical comparison yields insight into the cutoff fraction of the SMF. We argue that the similar columnar scales as well as the diversity of AERONET aerosol types shared by the two retrievals facilitates the analysis of their 2nd order intensive-parameter relationship. It is emphasized that the AERinv and SDA algorithms were developed independently and share no

explicit algorithmic links: what they do share is that the four spectral-AOD inputs to the AERinv represent 4 of the 5 spectral AODs employed as input to the SDA. What they do not share is the almucantar (angularly variable) radiance input employed in the AERinv retrieval. It is notable that the AERONET AOD is accurately measured with an estimated uncertainty of ~0.01 and 0.02 in the visible / NIR and the UV respectively (Eck et al., 1999) for an overhead sun (airmass or M = 1). This data quality enables both algorithms to, for example, retrieve notably consistent intensive parameter information such as particle

size. The consistency is also attributable to AERinv requiring a fit to the measured AOD spectra to within 0.01 while the 2nd order spectral fit to the AODs employed in the SDA has similar constraints for small AODs (see Figure 4 of O'Neill et all, 2001).

## 2 Theoretical considerations

### 2.1 Size-cutoff integration versus modal integration

Figure 1 illustrates the theoretical mechanical / optical framework of this paper (it is based on a simple lognormal fit to a sample AERinv particle-volume PSD retrieval[6]). That fit was applied to lend an air of empirical relevance to to this theoretical section: the discussion presented here is otherwise of a general nature and is not mean't to represent an algorithmic step of the AERinv retrievals (or of the SDA for that matter). The FM and CM PSDs are represented by the red and blue lognormal curves respectively. The SMF-type cutoff radius of $r_0$ is represented by the black dashed vertical line where the associated optical

and microphysical bimodal quantities are computed from both FM and CM PSDs to obtain sub-micron parameters to the left of $r_0$ and super-micron parameters to the right of $r_0$. The rest of the parameters in Figure 1 are defined immediately below as we develop the theoretical framework.

      Letting prime variables refer to integrations which are carried out over size regimes of $(0, r_0)$ and $(r_0, \infty)$ and unprimed variables refer to integrations carried out over entire modal features (over the entire FM or the entire CM PSDs) we can write

the total AOD $(\tau_a)$ at some given reference wavelength as the sum of the fine and coarse total-modal AODs;

$$\tau_a = \tau_f + \tau_c \qquad (1)$$

---

[6] The AERinv PSD particle-volume PSD ($dV/d\,lnr$) is readily transformed to the particle-surface PSD of Figure 1 ($dS/d\,lnr = 3/(4\,r)\,dV/d\,lnr$). See the Figure 1 caption for details.



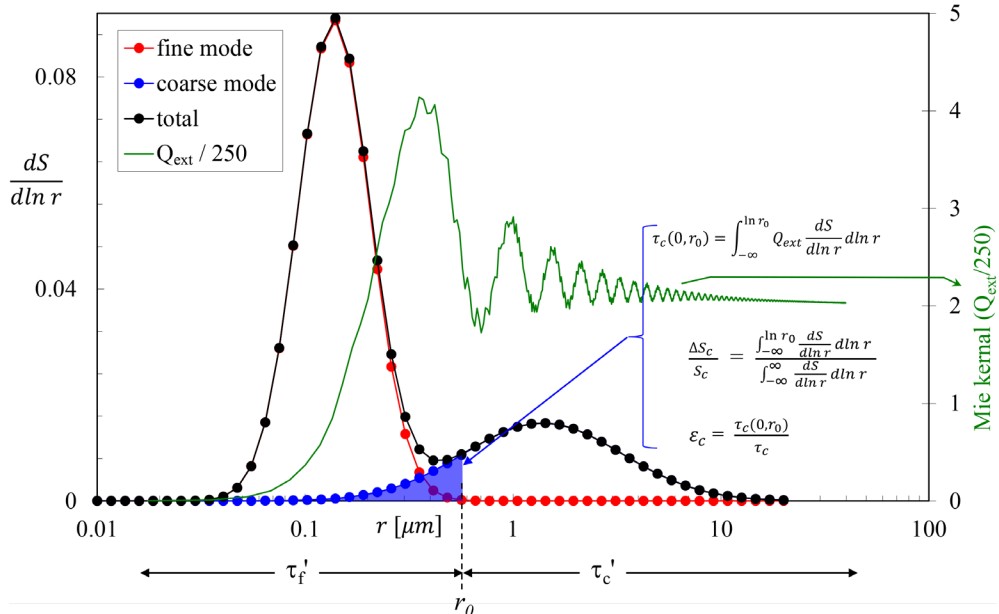

**Figure 1:** Illustrative, fine and coarse mode (red and blue) lognormal particle-surface size distributions ($dS/d\ln r$) and the Mie kernel (extinction efficiency $Q_{ext}$) for a wavelength of 0.5 μm and a refractive index of $1.44 - 0.0035i$ (by particle surface we mean the projected particle surface area of $\pi r^2$). The cutoff radius[7] ($r_0$) applies to a SMF type of computation. The total $dS/d\ln r$ (black curve) is (in order to have a level of grounding in reality) the sum of the FM and CM lognormal $dS/d\ln r$ curves that were individually manipulated to fit the mean of a series of AERONET and ground-based ($dV/d\ln r$) FM and CM sea-salt curves reported in Figure 8f of Reid et al. (2006). We base this illustration on $dS/d\ln r$ because it is more optically fundamental than the AERinv particle-volume PSD (the projected particle surface area is employed to normalize the extinction cross section to obtain the fundamental extinction efficiency). An analogous explanation (and a graph similar to that of Figure 2) would apply to the particle-volume PSD.

where (letting $\tau_x(r_1, r_2)$ refer to the optical integration from $r_1$ to $r_2$);

$$\tau_f = \tau_f(0, r_0) + \tau_f(r_0, \infty) \qquad (2a)$$

$$\tau_c = \tau_c(0, r_0) + \tau_c(r_0, \infty) \qquad (2b)$$

The analogous relationships in the domain of the cutoff size regimes are written;

$$\tau_a = \tau_f' + \tau_c' \qquad (3)$$

$$\tau_f' = \tau_f(0, r_0) + \tau_c(0, r_0) \qquad (4a)$$

$$\tau_c' = \tau_f(r_0, \infty) + \tau_c(r_0, \infty) \qquad (4b)$$

with an emphasis on the "conservation of $\tau_a$" ($\tau_f + \tau_c = \tau_f' + \tau_{fc}' = \tau_a$). The differences in the FM and CM (cutoff vs modal) integrations are respectively;

$$\tau_f' - \tau_f = \Delta\tau_f = \tau_f(0, r_0) + \tau_c(0, r_0) - [\tau_f(0, r_0) + \tau_f(r_0, \infty)]$$

$$= \tau_c(0, r_0) - \tau_f(r_0, \infty) \qquad (5a)$$

$$\tau_c' - \tau_c = \Delta\tau_c = \tau_f(r_0, \infty) + \tau_c(r_0, \infty) - [\tau_c(0, r_0) + \tau_c(r_0, \infty)]$$

---

[7] "inflection point" as it is called in the AERONET documentation.





$$= \tau_f(r_0, \infty) - \tau_c(0, r_0) \qquad (5b)$$

from which;

$$\Delta\tau_c = -\Delta\tau_f \qquad (5c)$$

The last expression is nothing more than confirmation of the "conservation of $\tau_a$".

**2.2 SMF versus FMF**

Using the definitions and relationships given above we can now define FMF and SMF respectively as;

$$\eta = \frac{\tau_f}{\tau_a} \quad \text{and} \quad \eta' = \frac{\tau_f'}{\tau_a} \qquad (6)$$

Given $\tau_f + \tau_c = \tau_f' + \tau_c'$ we divide by $\tau_a$ to obtain;

$$\eta + \frac{\tau_c}{\tau_a} = \eta' + \frac{\tau_c'}{\tau_a}$$

and, with a little manipulation;

$$\eta' = (1 - \varepsilon_c - \varepsilon_f)\eta + \varepsilon_c \qquad (7a)$$

where $\varepsilon_f$ and $\varepsilon_c$ represent the pure truncation errors of the FM and CM PSDs (the 2nd terms of equations 5(a) and 5(b) normalized by $\tau_a$);

$$\varepsilon_f = \frac{\tau_f(r_0, \infty)}{\tau_f} \quad \text{and} \quad \varepsilon_c = \frac{\tau_c(0, r_0)}{\tau_c} \qquad (7b)$$

These two quantities represent intensive parameter (largely quantity independent) attributes[8]. Typically however, $\varepsilon_c \gg \varepsilon_f$ (the major part of the FM PSD is well displaced to the left of $r_0$ as per the illustration of Figure 1). This has much more of a cutoff impact on the blue-coloured CM PSD of Figure 1 than on the red-coloured FM PSD: the solid blue cutoff portion ($\Delta S_c$) is a significant portion of the CM particle-surface density ($S_c$) and $\tau_c(0, r_0)$ is a significant optical depth portion of $\tau_c$ (i.e. both $\Delta S_c/S_c$ and $\varepsilon_c$ are typically significant) while the analogous FM fractions above $r_0$ are relatively insignificant. This affirmation, which we will empirically demonstrate in the multi-station analysis below, is, in part, related to the fact that the FM PSD is ~ half the width of the CM PSD (Dubovik et al., 2002: in their paper "width" specifically refers to the $\sigma$ value of fitted lognormal distributions).

Figure 2 is a plot of $\varepsilon_c$ vs $\Delta S_c/S_c$ for a variety of retrievals from the sites listed in Table 1. All these cases involved, as per the Figure 1 illustration, the fitting of FM and CM lognormal curves to AERinv particle-volume PSDs (and a subsequent transformation to particle-surface parameters). These lognormal fits permit the explicit (Mie-based) calculations of the equation (7b) fractions. They represent an (AERONET-grounded) theoretical illustration of an expected strong correlation between cutoff optics with cutoff mechanics (the correlation is less than monotonic because of variations in refractive index

---

[8] by "largely quantity independent" we mean, from an empirical standpoint, that $\tau_f(r_0, \infty)$ and $\tau_c(0, r_0)$ are typically well correlated with $\tau_f$ and $\tau_c$ respectively.

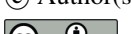



and the width of the lognormal curves for the different cases chosen in this illustration). We emphasize that automated lognormal fits to AERinv PSDs are not part of the empirical analysis process presented in this paper: rather the purpose of Figure 2 is to confirm the expected strong correlation between optical and microphysical CM cutoff fractions and thereby facilitate an understanding of $\varepsilon_c$'s role in the dynamics of equation (7a).

| Table 1: Site coordinates, predominant aerosol type, sampling periods and retrieval numbers for the AERONET inversions and the SDA retrievals employed in this study | | | | | | | |
|---|---|---|---|---|---|---|---|
| Sites | Latitude | Longitude | Elevation (m) | Region | Aerosol class / type | Sampling period | Number of retrievals[a] |
| GSFC | 38° 59' 33" N | 76° 50' 23" W | 87 | Maryland, USA | Urban-industrial | 1994-2020 | 13070 |
| Mongu | 15° 15' 13" S | 23° 9' 3" E | 1047 | Zambia | Biomass Burning | 1997-2009 | 4751 |
| Hamim | 22° 58' 0" N | 54° 18' 0" E | 105 | United Arab Emirates | Dust | 2004-2007 | 2295 |
| Solar Village | 24° 54' 25" N | 46° 23' 50" E | 764 | Saudi Arabia | Dust | 1999-2012 | 14171 |
| Forth Crete | 35° 19' 58" N | 25° 16' 57" E | 20 | Greece | Maritime and Dust | 2003-2017 | 4039 |
| Midway Island | 28° 12' 35" N | 177° 22' 42" W | 20 | Midway Island | Maritime | 2001-2014 | 644 |
| Lanai | 20° 44' 6" N | 156° 55' 18" W | 20 | Hawaii | Maritime | 1997-2004 | 1175 |
| PEARL[b] | 80° 3' 13"N | 86° 25' 1" W | 615 | Nunavut, Canada | Arctic | 2007-2019 | 270 |
| Barrow | 71° 18' 44"N | 156° 39' 54" W | 8 | Alaska, USA | Arctic | 1997-2020 | 351 |
| Thule | 76° 30' 58" N | 68° 46' 8" W | 225 | Northern Greenland | Arctic | 2007-2019 | 500 |
| [a] See the methodology section of the text for details on how the AERONET inversion and SDA retrievals were matched. [b] Polar Environment Arctic Research Lab. The acronym represents the total atmospheric research infrastructure at Eureka, Nunavut. The AERONET/AEROCAN site is more accurately referred to as the PEARL Ridge Lab. | | | | | | | |

The linear form of equation (7a) with its sub-unity slope of $(1 - \varepsilon_c - \varepsilon_f)$ and positive $\varepsilon_c$ intercept is coherent with

empirical results of $\eta'$ being generally $>\sim \eta$ (the well recognized empirical result, supported by the results presented below, of SMF being generally $>\sim$ FMF). Because $\varepsilon_c$ generally dominates $\varepsilon_f$ the linear $\eta'$ vs $\eta$ relation pivots clockwise about the point $(\eta,\ \eta') = (1,\ 1)$ in a "linearly inverse" fashion (slope and intercept of approximately $1 - \varepsilon_c$ and $\varepsilon_c$). Figure 3 shows a set of equation (7a) straight lines for representative ranges of different $\varepsilon_c$ and $\varepsilon_f$ values that were inspired by the ranges encountered in the empirical results that follow.

Cases of large $\varepsilon_f$ are relatively rare but would occur during intense FM events characterized by large-amplitude FM PSDs that are shifted towards larger radii: illustrations include strong (large FM AOD) smoke events (see, for example, the sub-Arctic (Alaskan) smoke cases of AOD(440 nm) $>\sim 1$ in Figure 9a of Eck et al., 2009 and strong FM pollution events enhanced by high relative humidity (see, for example, the PSDs corresponding to AOD(440 nm) $>\sim 2$ in Figure 4 of Eck et al., 2020). However an increasing $\varepsilon_f$ presupposes that $r_0$ is fixed (as for volumetric surface sampling devices): the AERinv

technique of setting the $r_0$ value to the minimum of the PSD (see below) results in a minimization of $\varepsilon_f$.



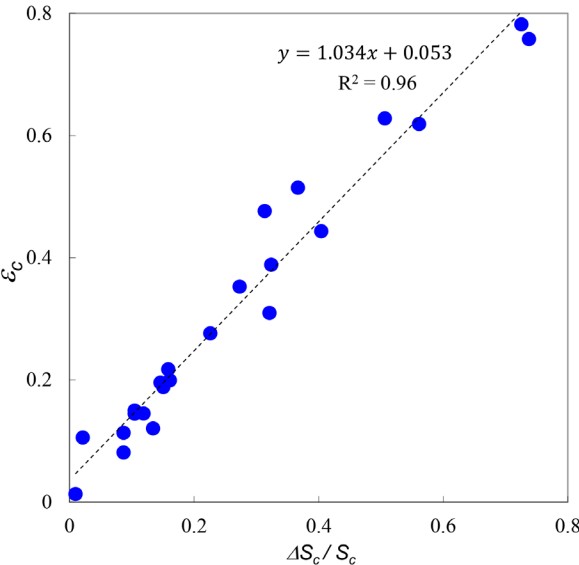

**Figure 2:** Plot of $\varepsilon_c$ vs $\Delta S_c / S_c$ for a variety of simulated optical depth retrievals. The lognormal FM and CM PSDs employed in these simulations were obtained by fitting their sum to retrieved AERinv PSDs over a variety of sites whose predominant aerosol type was urban, dust or marine (and employing the refractive indices provided by the retrieval product). The AERinv PSD retrievals were Version 2 (for which $r_0$ was fixed at 0.6 µm and, accordingly, for which the $\varepsilon_c$ vs $\Delta S_c / S_c$ regression would be only dependent on $\varepsilon_c$ and 2$^{nd}$ order parameters such as refractive index (i.e. independent of $r_0$).

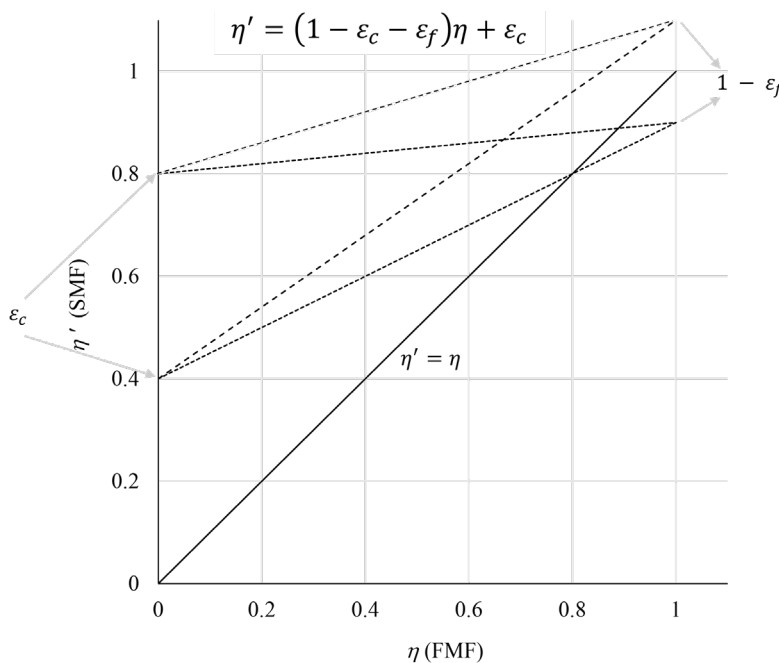

**Figure 3:** $\eta'$ (SMF) vs $\eta$ (FMF) lines of equation (7a) for three $\varepsilon_c$ values and three $\varepsilon_f$ values (0, 0.4, 0.8 and -0.1, 0 and 0.1 respectively). The $\varepsilon_f$ values were inspired by the range of empirical values seen in Figure 6 (and, in the case of $\varepsilon_f$ by the range of values, whether real or artefactual, seen in Figure 7).



## 3 Methodology

Hesaraki et al. (2017) give an overview of the AERinv and SDA with an emphasis on the fact that the former is a significantly more comprehensive algorithm whose low frequency sampling rate is appropriate for detailed climatological scale analyses (see, for example, Dubovik et al., 2002 and AboEl-Fetouh et al., 2020) while the high frequency sampling rate of the SDA is better suited to the detailed analysis of diurnal events (see, for example, Figure S5 of Saha et al., 2010). The SDA is readily applied to AOD spectra generated by starphotometers and moonphotometers (see, for example, Baibakov et al., 2015) while analogous AERinv products (requiring an almucantar scan) do not exist for nighttime conditions.

In this investigation we "matched" SDA to AERinv retrievals by employing averages of Version 3 Level 2.0 AODs (Giles et al., 2019) as inputs to the SDA if those AODs were within a time window of $\pm$ 16 minutes about the nominal AERinv times. Version 3 AERinv products (Sinyuk et al., 2020) were employed to derive estimates of SMF, $\tau_f'$ and $\tau_c'$. The cutoff radius for those products is actually defined as the minimum of the AERinv output particle-volume PSD ($dV/d\ln r$) with a restriction that the radius bin centers of that minimum must be one of four AERinv bins ($r_0$, which is referred to as an "inflection point" in AERONET documentation is allocated a bin-center value of 0.439, 0.576, 0.756 or 0.992 μm). The AERinv retrievals of $\tau_f'$ and $\tau_c'$ are interpolated to the SDA reference wavelength of 500 nm using a 2nd order (log-log) space spectral polynomial regressed to the $\tau_f'$ and $\tau_c$ values at the four AERinv wavelengths of 440, 675, 870 and 1020 nm (the same technique employed in the SDA).

A variety of AERONET sites (Figure 4), representing different types (classes) of aerosols were chosen to investigate the SMF versus FMF relationship. These aerosol classes (Table 1) included sites known for FM urban-industrial aerosols (GSFC in Greenbelt MD), FM biomass burning (Mongu, Zambia), CM dust (Crete, Hamim and Solar Village), CM maritime aerosols (Midway Island and and Lanai), a mixture of dust and marine aerosols (Forth Crete) and a mixture of high-Arctic aerosols (PEARL and Thule) as well as low-Arctic aerosols (Barrow).

The Arctic category is a mixed class of aerosols: its inclusion in our table of aerosol types is more in terms of it representing an important comparative test (relative to the applicability of southern latitude findings) in a region where the aerosol signal is notably weak. Arctic illustrations of FMF vs SMF principles feature more in the analysis presented below precisely because the signal is weak: the observation of a specific FMF vs SMF trend or characteristic in the Arctic is an indicator of the robustness of that observation. This weak-signal robustness is something we often see for Arctic retrievals (see, for example, the FM and CM results of AboEl-Fetouh et al., 2020): it may well be (at least in part) attributable to the large solar zenith angles (large M) and attendant 1/M decrease in optical depth errors (a typical finding over 20 years of AERONET Mauna Loa calibrations: see also Figure 2 of Karanikolas et al., 2022 for an empirical validation of this 1/M error dependency).





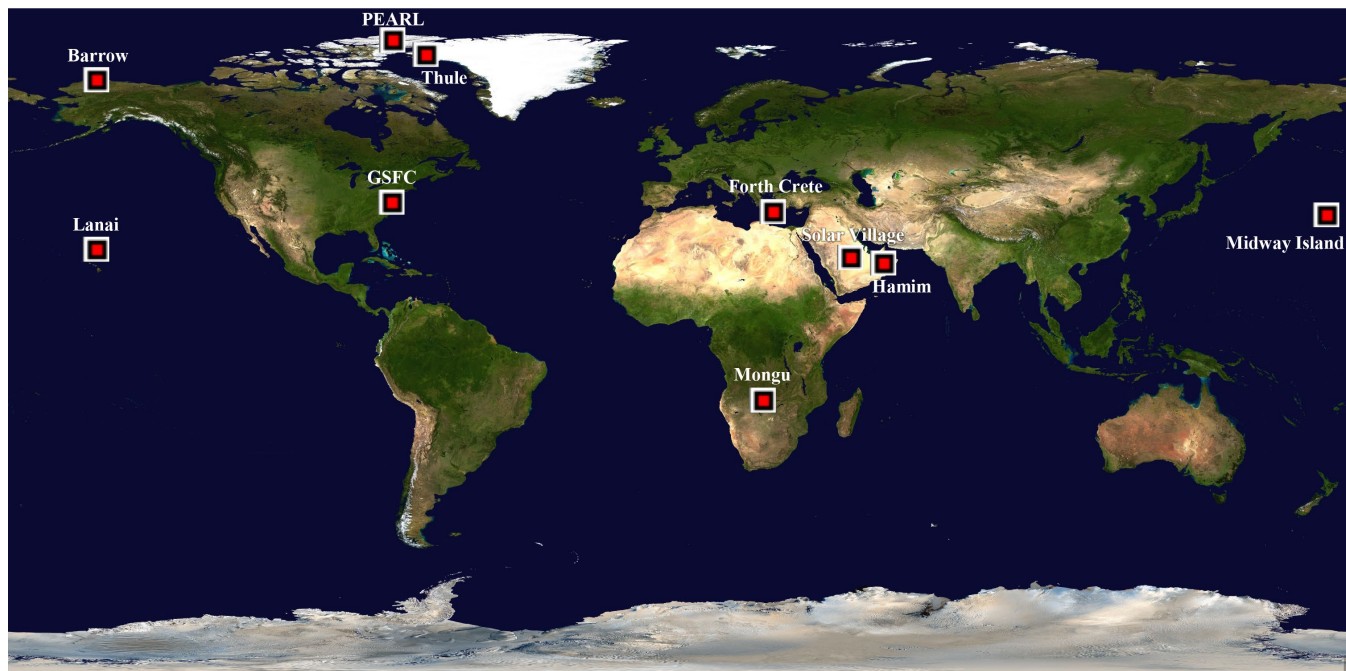

**Figure 4:** AERONET stations employed for generating the statistics employed in this paper. These stations were chosen to represent a regional variety of aerosols (pollution, biomass burning, dust mixtures, sea-salt mixtures and Arctic aerosols). See Table 1 for details.

## 4 Results

### 4.1 Frequency of Occurrence of inflection ($r_0$) points

Figure 5 shows the relative frequency of occurrence (FO) of the four different AERinv inflection ($r_0$) points for the different Table 1 classes. The relative importance of CM PSDs vs FM PSDs for the dust class sites of Solar Village and Hamim and the marine sites of Midway Island and Lanai push the AERinv minimum to smaller $r_0$ values (resulting in an FO dominance at the 0.439 μm inflection point). The marine sites are moderately less asymmetric (less pushed towards 0.439 μm) than the dust sites because the CM PSDs for the dust sites tend to be of larger amplitude. There are, however, other factors at play that could have some impact on the FO distributions: for example, large FM sulfatic particles from Kiluea eruptions that might push back on the CM PSD dominance of marine particles at Lanai and/or non sphericity effects of dust that, when corrected (Dubovik et al., 2006) would produce a significantly larger FM PSD amplitude (a complication that does not impact the predominantly spherical marine particles).

The two sites that are heavily influenced by strong FM PSDs (the Mongu site and the GSFC urban-industrial site and) tend (as suggested above in an $\varepsilon_f$ context) to "push" the PSD minimum towards larger $r_0$ values relative to the dust and marine sites (resulting in a more balanced FO curve in Figure 5). Eck et al. (2001) attributed this FM particle-size increase to coagulative effects for Mongu (see their Figure 6) while Eck et al. (2012) attributed the increase to the effects of





hygroscopically induced FM particle growth at the GSFC site (see their Figure 17). Other smoke impacted and urban industrial

sites show similar coagulative particle growth effects (see Figure 10b of Eck et al., 2019 for specific cases recorded in the

southeast Asian tropical forest) and hygroscopic particle growth impacts (see, for example Figure 13 of Eck et al., 2005 for

Beijing). We argue below that these large amplitude increases in FM particle size are, given the AERinv technique of variable

inflection points, relatively minor in terms of producing significant $\varepsilon_f$ values.

The Arctic sites show a FM PSD domination that produces an FO distribution that is not unlike the Mongu distribution for

Barrow while being much more skewed towards large $r_0$ for PEARL and Thule. We illustrate below (as part of a discussion

on the variation of $\varepsilon_c$) that a systematic (seasonal), spring to summer inflection point increase can be attributed to the Arctic

sites.

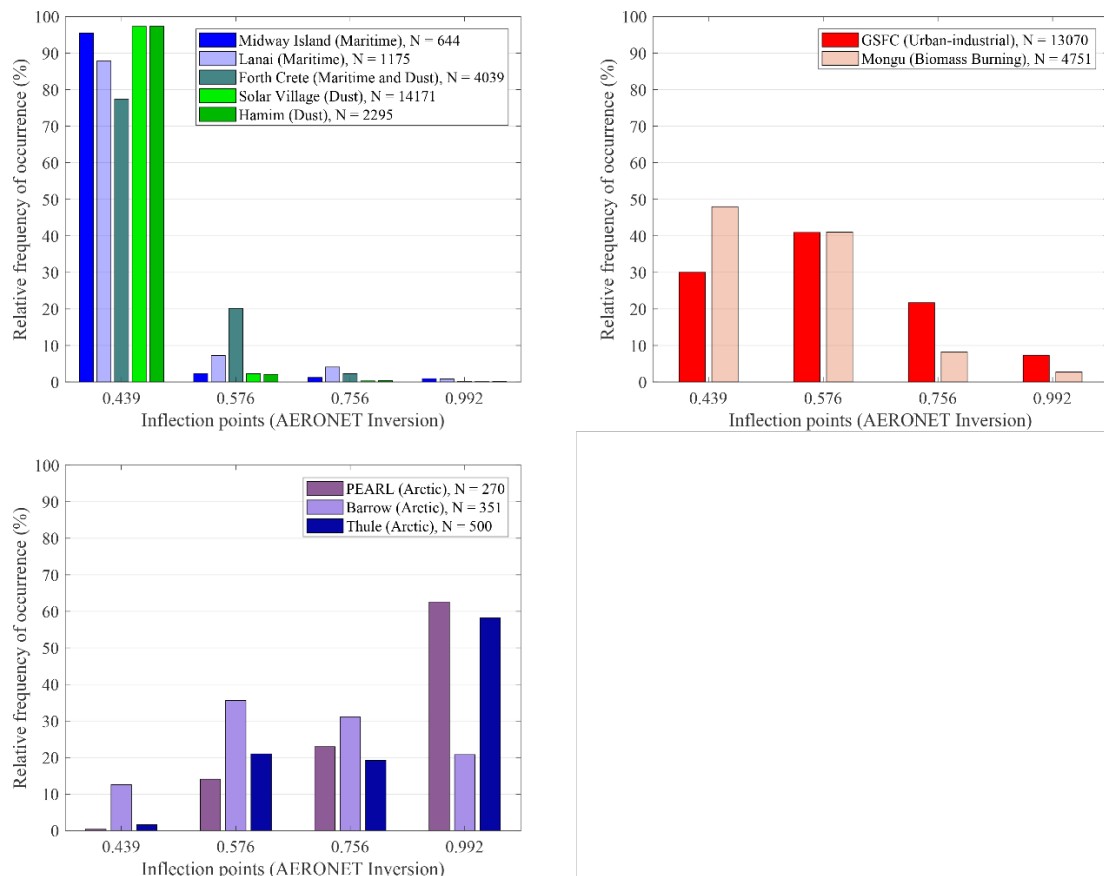

**Figure 5:** Relative frequency occurrence (FO) distributions for the four different AERONET inflection points ($r_0$ values) for all Table 1
AERONET sites.

## 4.2 $\eta'$ vs $\eta$ (SMF vs FMF) scattergrams

Figure 6 shows $\eta'$ vs $\eta$ (SMF vs FMF) scattergrams representing four key aerosol types of Table 1 (scattergrams for

the rest of the aerosol types and sites can be seen in Figure S1). The theoretical solid black lines of Figure 6 represent various



values of $\varepsilon_c$ in equation (7a): we chose to set $\varepsilon_f$ to zero in tracing those lines because its value is, as indicated above, generally

small (and to not obscure the graphs with 2nd order detail). We note (as per the discussion of the $r_0$ FO curves) that as the AERinv PSD minimum increases, the cutoff portion of the CM PSD increases (resulting in the transition from red to blue curves). The slopes tend to decrease (swing clockwise) with an attendant increase in the intercepts.

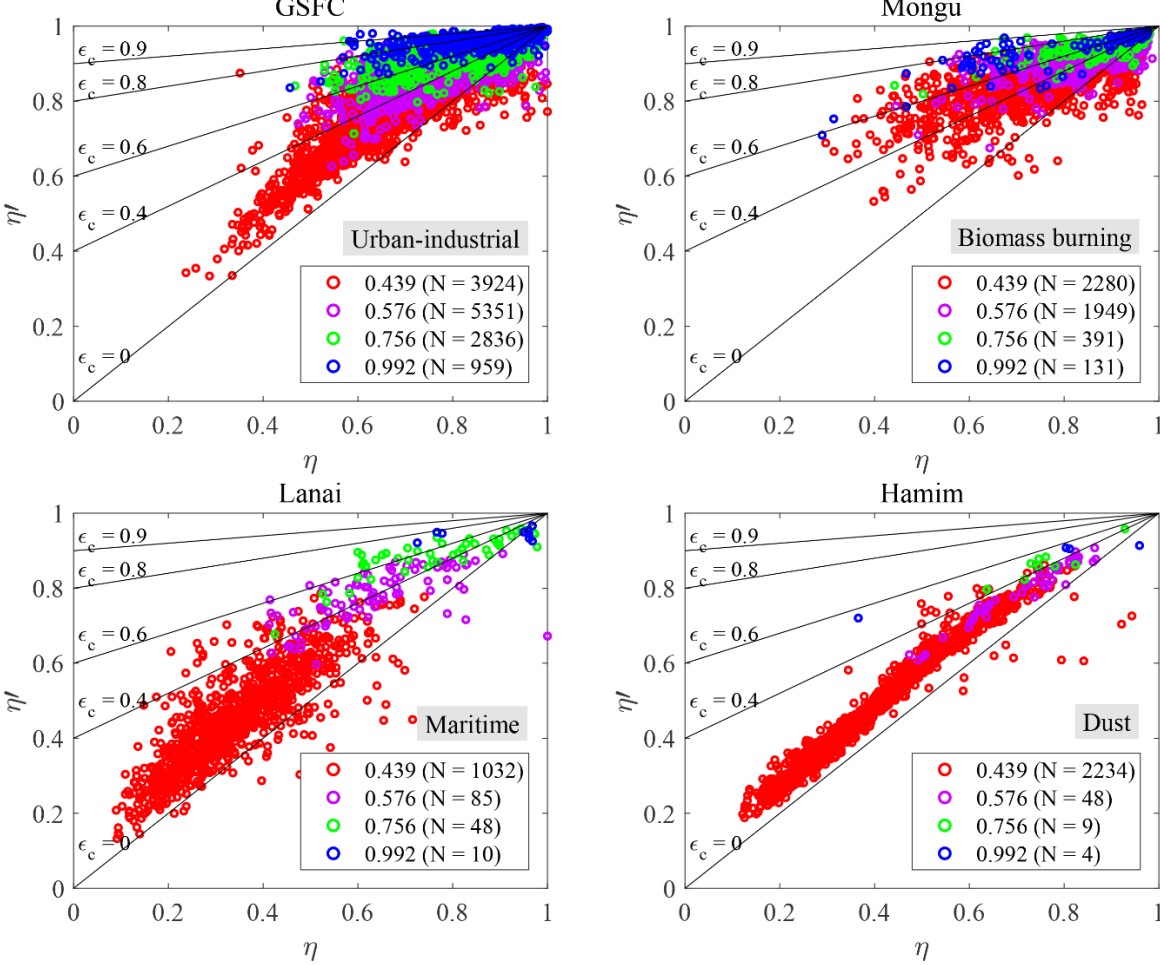

**Figure 6:** $\eta'$ vs $\eta$ (AERinv-derived SMF vs SDA-derived FMF) scattergrams for the sites of GSFC, Mongu, Lanai, and Hamim
(respectively aerosol types of urban-industrial, biomass burning, marine and dust as per Table 1). The remaining scattergrams for the other sites are shown in Figure S1. The solid black lines are those of equation (7a) with (for the sake of the simplicity of the presentation coupled with the fact that $\varepsilon_f$ has a relatively minor role), $\varepsilon_f$ was set to 0.

The FO dominance of the 0.439 μm $r_0$ bin is most evident for the CM dominated dust class (Hamim) with, practically

speaking, a single (small $\varepsilon_c$) red-coloured grouping of points being observable while the more balanced distributions of FO

curves for GSFC and Mongu, provide four clear groupings of points. In the extreme of CM cutoff, virtually all of the optically

significant contributions of the CM PSD are cut off and an asymptote of $\eta' \to 1$ with $\varepsilon_c \to 1$ for all values of $\eta$ is

approached (the slope of equation (7a) approaches zero). That extreme CM cutoff condition is most evident for the blue, large-



$r_0$ points of the GSFC scattergram) where a regression slope could be almost parallel to the $\eta'$ axis. We did indeed find $r_0 =$ 0.992 cases associated with near unity $\eta'$ values for which most of the optically significant portion of the CM PSD was cut

off while the FM and CM PSDs were not inordinately unbalanced in terms of amplitude ($\eta$ was some sub-unity value of significance for which there were no apparent problems associated with the AOD spectra).

The small to large $r_0$ (red to blue) transformation translates, for example, into a classical seasonal pattern for the Arctic scattergrams of Figure S1: the springtime amplitude of the small-sized CM PSD (that AboEl-Fetouh et al., 2020, associated with Asian dust) decreases progressively from spring to summer. This weakening of CM influence induces a spring to summer

increase in the value of $r_0$ and, by extension, $\varepsilon_c$ (see the seasonal $r_0$ histograms and derived table for Barrow in Figure S2). This is an effect that is more noticeable in the Level 1.5 AERinv products of ibid than the Level 2.0 products of this paper[9].

Within the theoretical context of equation (7a), the scattergrams of Figure 6 show (predominantly for the red-coloured 0.439 μm inflection value of $r_0$), a minority of unphysical points below the $\eta' = \eta$ line (points for which $\eta' < \eta$). An investigation into the most extreme cases of this inequality indicated that the wayward points were more inclined to be

associated with abnormally large values of $\eta$ (rather than abnormally small values of $\eta'$) and small AODs (<~ 0.05). Figure 10 of O'Neill et al. (2003) shows the noise sensitivity of $\eta$ values to such small AODs if one assumes an RMS AOD error of 0.01 in all bands (0.01 being <~ errors of AERONET field instruments). In general, this type of small-AOD sensitivity was reported empirically by, for example, Eck et al. (1999) and O'Neill et al. (2000). The fundamental dynamic is that a band-to-band AOD discontinuity <~ 0.01 can generate important perturbations in small-AOD curvature spectra and produce significant

outliers in the spectrally sensitive $\eta$ values.

**4.3 Analysis of the slopes and intercepts (derivation of $\varepsilon_c$ and $\varepsilon_f$)**

The $\eta'$ vs $\eta$ scattergrams support the hypothesis that there is a physical / optical interpretation that can be given to the slope and intercept of equation (7a) (that equations (7a) and (7b) are theoretically relevant approximations). Figure 7a and 7b show the regression slopes vs the intercepts derived for all sites and all $r_0$ values of the Figure 6 and supplementary material

scattergrams (with a coherent color scheme between the scattergrams and Figure 7a) while Table 2 lists the associated regression statistics for the 0.439 μm inflection point. The large range of the intercept variation seen in Figure 7 confirms that the cutoff portion of the CM PSD ($\varepsilon_c$ of equation 7b) is much more determinant in affecting the interplay of $\eta'$ vs $\eta$. The position of the colored circles relative to the dashed or dotted black lines (derived from equations (7a) and 7(b)) visually support the Table 2 results of small $\varepsilon_f$ ($\varepsilon_f$ values being <~ 0.07 according to their positions between the $\varepsilon_f = 0$ and $\varepsilon_f =$

0.1 grid lines of Figure 7).

---

[9] the Level 2.0 processing tends to eliminate springtime retrievals completely: the majority of eliminations are due to excessive residuals in the retrieved vs measured sky radiances that are, in turn, largely incited by the strong reflectance uncertainty of springtime snow (and its attendant impact on computed sky radiance) as well as the AERinv protocol of eliminating Level 2 retrievals if any snow is detected (by MODIS) within a 5 km radius of the site.





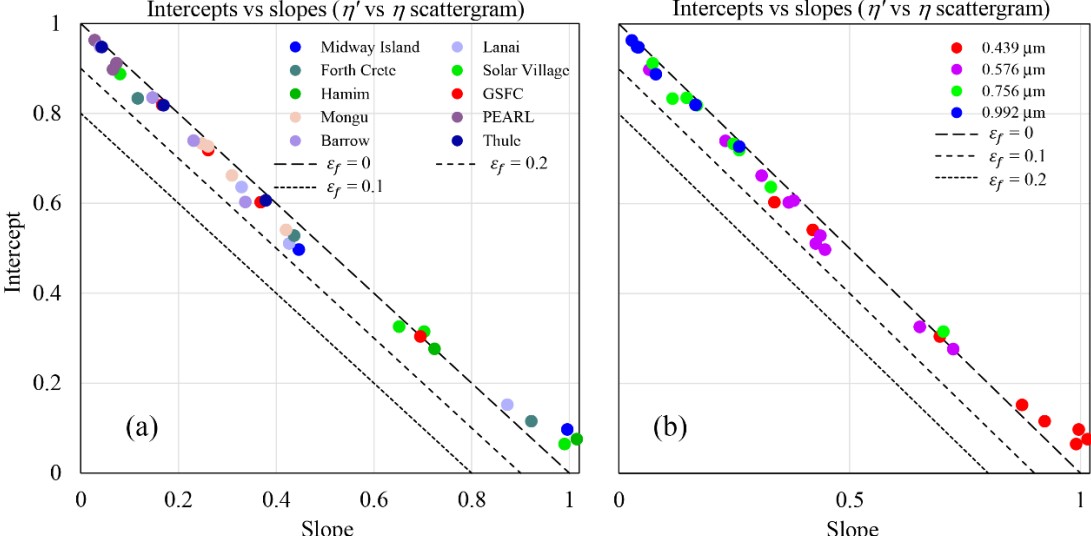

**Figure 7:** Intercept vs slope plots for the $\eta'$ vs $\eta$ scattergrams as a function of (a) the cutoff $r_0$ value (AERinv inflection point radius) and (b) all the sites of Table 1. The black dashed or pointed lines represent the family of straight lines generated by equation (7a): Slope $= \left(1 - \varepsilon_c - \varepsilon_f\right)$ and Intercept $= \varepsilon_c$ for values of $\varepsilon_c = 0$ to 1 and three values of $\varepsilon_f$ (0.0, 0.1 and 0.2).

The values corresponding to the four red-colored (0.439 μm inflection point) circles for the desert and marine sites (bottom right hand corner of Figure 7b) are however outside the physically coherent range of equation (7). The negative values of $\varepsilon_f$ (see Table 2) can be inferred from the $\varepsilon_f = 0, 0.1, 0.2$ grid of Figure 7b or, more visually, from the unphysical (super-unity) slopes that arise from a regression through the red points of the desert and marine scattergrams of Figure 6. This incoherency is very likely the result of those regression lines representing a non-unique ensemble of $\varepsilon_c$ and $\varepsilon_f$ values. From

an opto-physical standpoint, they vary as a function of the diversity of dust and marine CM PSDs that are predominantly associated with the 0.439 μm inflection point (they roughly lie between the $\varepsilon_c = 0$ and $\varepsilon_c = 0.4$ boundaries of Figure 3: the $\eta = 1$ intercept of the expression $(1 - \varepsilon_f)$ will (in consequence of a super-unity slope) yield negative values above the $\eta' = \eta$ intercept).

Non-systematic noise will also impact the derived $\varepsilon_c$ and $\varepsilon_f$ values. Scattergrams of $\tau'_f$ vs $\tau_f$ for the two sites that are

largely dominated by strong FM variations (GSFC and Mongu) and which showed the most balanced (Figure 5) FO distributions are shown in Figure A1. One can observe, in the first instance, the point-dispersion reduction and the convergence towards the $\tau'_f = \tau_f$ line as both increase (as $\tau_a$ increases). The extensive (quantity-dependent) nature of those scattergrams complements the analogous $\eta'$ vs $\eta$ semi-intensive scattergrams of Figure 6 by explicitly displaying the noise-like influences of the amount of aerosol ($\tau'_f$ or $\tau_f$) as well as $r_0$. The $\tau'_f$ vs $\tau_f$ dispersion of Figure A1 is nonetheless generally small: this

underscores a hypothesis that the $\eta'$ vs $\eta$ results presented above are robust 2nd order findings. The FO distributions of $\eta'$ vs $\eta$ in Figure A2 effectively eliminate the extensive variations of Figure A1 with an attendant enhancement of the 2nd order influences.





| Table 2: $\eta'$ vs $\eta$ regression stats for $r_0 = 0.439$ µm [a] | | | | | | | | | | | |
|---|---|---|---|---|---|---|---|---|---|---|---|
| | m | b | $R^2$ | $\sigma(\eta'_{res})$ | $\sigma(m)$ | $\sigma(b)$ | N | $\varepsilon_c$ | $\varepsilon_f$ | $\sigma(\varepsilon_c)$ | $\sigma(\varepsilon_f)$ |
| GSFC | 0.695 | 0.304 | 0.749 | 0.043 | 0.409 | 0.288 | 3924 | 0.304 | 9E-04 | 0.288 | 0.29 |
| Mongu | 0.42 | 0.541 | 0.641 | 0.038 | 0.314 | 0.256 | 2280 | 0.541 | 0.039 | 0.256 | 0.182 |
| Hamim | 1.015 | 0.076 | 0.969 | 0.025 | 0.172 | 0.069 | 2234 | 0.076 | -0.091 | 0.069 | 0.158 |
| Solar Village | 0.99 | 0.065 | 0.912 | 0.043 | 0.314 | 0.12 | 13790 | 0.065 | -0.055 | 0.12 | 0.291 |
| Forth Crete | 0.922 | 0.116 | 0.941 | 0.044 | 0.226 | 0.139 | 3125 | 0.116 | -0.038 | 0.139 | 0.178 |
| Midway | 0.996 | 0.097 | 0.832 | 0.059 | 0.39 | 0.15 | 615 | 0.097 | -0.093 | 0.15 | 0.36 |
| Lanai | 0.873 | 0.152 | 0.674 | 0.072 | 0.427 | 0.154 | 1032 | 0.152 | -0.025 | 0.154 | 0.398 |
| PEARL | N/A (N = 1) | | | | | | | | | | |
| Barrow | 0.34 | 0.604 | 0.447 | 0.053 | 0.462 | 0.341 | 44 | 0.604 | 0.057 | 0.341 | 0.311 |
| Thule | 0.259 | 0.66 | 0.196 | 0.038 | 0.482 | 0.405 | 8 | 0.66 | 0.081 | 0.405 | 0.262 |

[a] See for example, Taylor (1997) for typical regression relationships. The $\varepsilon_c$ and $\varepsilon_f$ values as well as their standard deviation ($\sigma(\varepsilon_c)$ and $\sigma(\varepsilon_f)$) were derived from the regressed slope and intercept ("m" and "b") using the slope and intercept expressions of equation (7a). The values of ($\sigma(\varepsilon_c)$ and $\sigma(\varepsilon_f)$) are effective standard deviations computed from effective standard deviations of $m$ and $b$. By this we mean that the standard error (from sources such as Taylor, 1997) is multiplied by $\sqrt{N}$ to yield $\sigma(m)$ and $\sigma(b)$: this transforms the unrealistically small $m$ and $b$ uncertainties into values that are more representative of the variation seen in the scattergrams of Figure 6. This change is coherent with the notion that those variations are more likely due to $\varepsilon_c$ and $\varepsilon_f$ being characterized by a systematic range of values for any given inflection point ($r_0$) value. Note that $\sigma(\eta'_r)$ represents the standard deviation of the regression residuals (res).

The Figure 8 FO distributions of $\eta' - \eta$ vs $\tau_a$ more readily show the decreasing dispersion and the convergence towards the $\eta' = \eta$ ($\tau'_f = \tau_f$) line with increasing $\tau_a$ (increasing $\tau_f$ values for GSFC and Mongu). The broad left to right movement of the (red) FO peak values with increasing $r_0$ (notably in the case of GSFC) also clarifies an aspect that is not easily discernable in the highly correlated scattergrams of Figure A1: that an increase in $r_0$ is coarsely associated with $\eta'$ values that approach $\eta$ or hence, $\tau'_f$ values that approach $\tau_f$ (a trend that effectively drives the $\eta' = \eta$ ($\tau'_f = \tau_f$) convergence incited by increasing $\tau_a$).

Figure 9 shows the regression-derived $\varepsilon_c$ and $\varepsilon_f$ variation as a function of an artificial minimum ($\tau_{a,min}$) in the lower bound of the $\tau_a$ regression range (see the Figure 9 caption for more details). The result shows no strong $\varepsilon_c$ dependency on $\tau_{a,min}$ and more variable but consistently small amplitudes of $\varepsilon_f$)[10]. Qualitatively, this observation is not unexpected given the fairly persistent slope of the high FO (red-colored) ellipses of Figure A2).

---

[10] except in the case of the (red) 0.439 µm $r_0$ (yellow-filled) value where the $\varepsilon_f$ variations were more substantial. At the same time, the number of regressions points ($N$) rapidly decreases with increasing $\tau_{a,min}$ so that large $\varepsilon_f$ (and $\varepsilon_c$) deviations are less statistically significant.



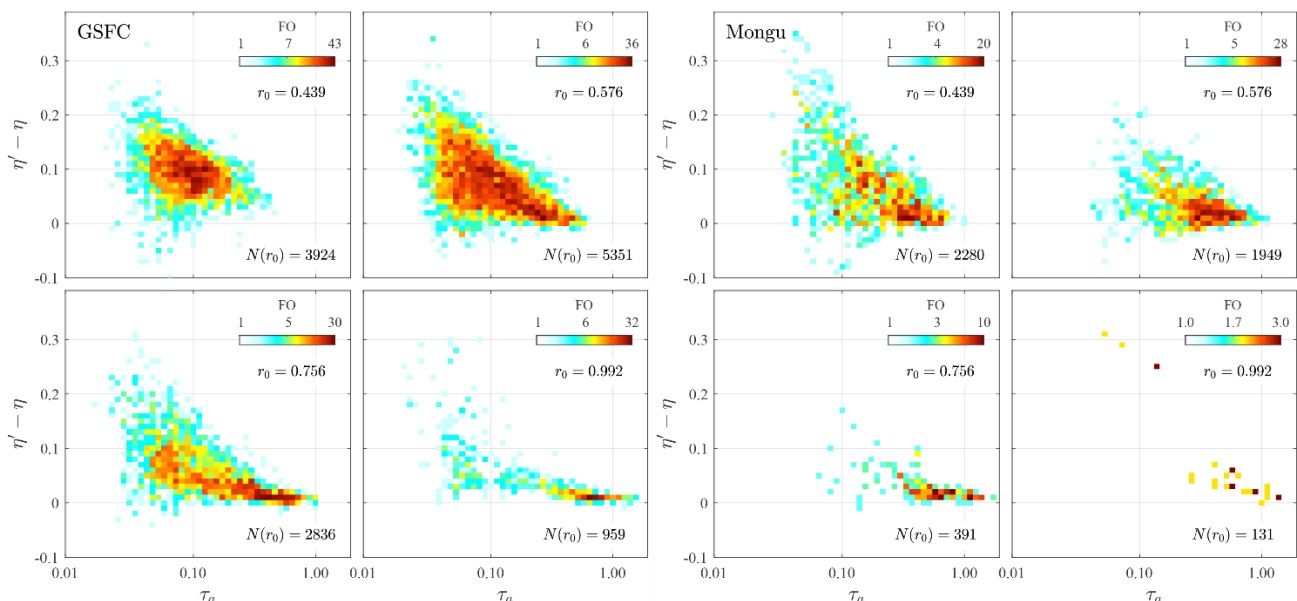


**Figure 8:** $\eta' - \eta$ vs $\tau_a$ FO distributions for GSFC and Mongu.

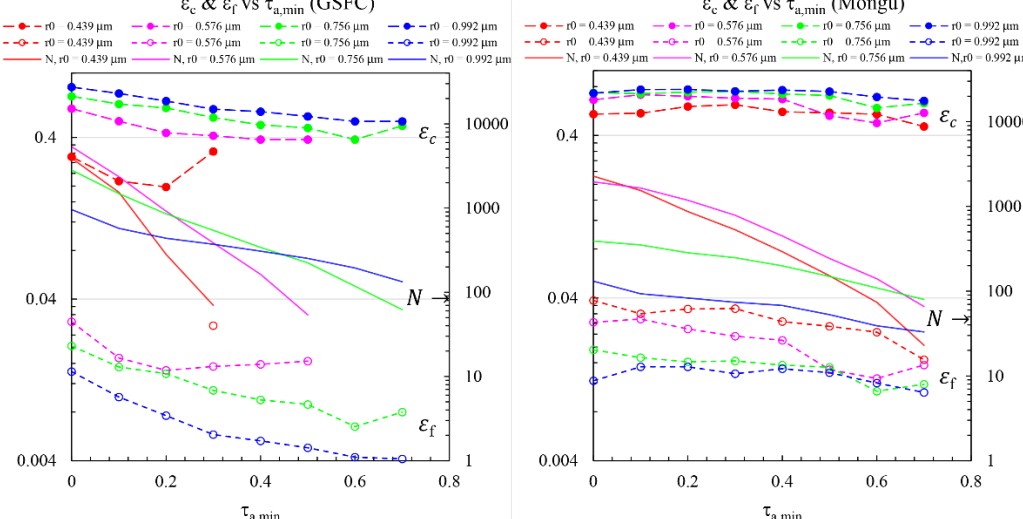

**Figure 9:** GSFC and Mongu variations of regression-derived values of $\varepsilon_c$ and $\varepsilon_f$ as a function of a minimum value of total AOD at 500 nm ($\tau_{a,min}$) employed in the $\eta'$ vs $\eta$ regressions[11] (to be clear, the regressions of Figure 6 and the regression-derived $\varepsilon_c$ and $\varepsilon_f$ values of Figure 7 were obtained with no restrictions on the matched AERinv and SDA retrievals). Same colour scheme as Figure 6 and 7. The number of
matched retrievals ($N$) are also shown (right-hand axis). Result are not shown in cases where $N < 20$. The orange filled large circle represents the only $\varepsilon_f$ point of the $r_0 = 0.439$ μm regressions that survived the $N < 20$ criterion and that did not yield a negative $\varepsilon_f$ value.

---

[11] The decreasing point dispersion (the approach of $\tau_f'$ to $\tau_f$) with increasing $\tau_f$ indicates that the regression-derived precision of the $\varepsilon_c$ and $\varepsilon_f$ values should, for that reason alone, generally increase with larger values of $\tau_f$. However, offsetting this type of influence on an increase in precision is the attendant decrease in $\tau_f$, the decrease in the number of regression points and the squeeze in $\tau_f$ and $\tau_f'$ variability as the $\eta' = \eta$ singularity is approached.



The choice of a variable $r_0$ in the AERinv retrievals clearly means that the affirmation of increasing $\varepsilon_f$ in the presence of strong FM events (what would be measured by a traditional fixed-$r_0$ device) cannot readily be observed with the AERinv

retrievals. Strong FM events basically push the FM PSD to larger radii (while the CM PSD remains relatively inactive[12]). The AERinv approach of selecting $r_0$ as the minimum of the PSD basically neuters the cutoff of significant optical portions of the FM PSD. The general tendency for $r_0$ to increase with increasing $\tau_f$ in Figure 8 is consistent with the concept of the FM PSD being pushed to the right as $\tau'_f$ and $\tau_f$ increase.

AERinv PSDs can differ from the simple bi-modal paradigm incorporated in equations (1) through (7). The bi-modality

of the Arctic CM PSDs of AboEl-Fetouh et al. (2020) were ascribed to a small-sized CM feature in the 1.3 μm AERinv bin (a systematic feature that they attributed to Asian dust) and a larger sized mode that might have been linked to local dust and/or sea-salt. Eck et al. (2012) reported a bi-modal FM PSD (their Figure 3) that they attributed to cloud processing of FM pollution (haze) aerosols. In such cases one can appeal to the optical equivalency of, for example, a bi-modal CM PSD to a single CM PSD whose parameters become curvature-parameter averages of the two CM components (see Appendix A for the two specific

cases of a bimodal CM PSD with a unimodal FM PSD as well as a a bimodal FM PSD with a unimodal CM PSD). This means the bi-modal expressions of O'Neill et al. (2001) still apply for the AERONET SDA product and thus that the FMF to SMF expressions (equations (1) to (7)) can still be used.

The FMF (SDA) approach is arguably the more fundamental approach for separating FM and CM optical contributions because it is intrinsically related to the modal nature of different types of aerosols (at least to the first order: modality becomes

obscured, for example, with internal mixing of different aerosol types). The SMF (AERinv) approach is the more pragmatic as it is commonly and readily applied to microphysical surface and airborne measurements. It is however, as we have seen in results like those of Figure 6, very dependent on the selected cutoff radius.

## 5 Conclusions

We presented a simple SMF vs FMF ($\eta'$ vs $\eta$) equation that enabled an understanding of the well recognized empirical

result of SMF being larger than the FMF . This result has been reported for in-situ, satellite and ground-based remote sensing techniques : our focus was on an SMF vs FMF interpretation in the form of AERinv SMF vs SDA FMF retrievals. We pointed out that these two AERONET products provide a unique opportunity to empirically compare the SMF and FMF approaches at similar (columnar) remote sensing scales and across a shared global variety of aerosol types.

The SMF vs FMF equation largely captured the SMF vs FMF behavior of the AERinv vs SDA products as a function

of inflection point ($r_0$) across an ensemble of AERONET sites and aerosol types (urban industrial, biomass burning, dust,

---

[12] and suffers from a per-particle extinction kernel (the green-colored $Q_{ext}$ factor of Figure 1) that is weaker in CM radius range.





marine, maritime and Arctic). The SMF vs FMF behavior was primarily dependent on the intensive parameter of relative cutoff portion of the CM PSD ($\varepsilon_c$) and, to a second order the relative cutoff portion of the FM PSD ($\varepsilon_f$). The overarching dynamic was that the linear *SMF* vs *FMF* relation pivots clockwise about the point ($\eta$, $\eta'$) = (1, 1) in a "linearly inverse" fashion (slope and intercept of approximately $1 - \varepsilon_c$ and $\varepsilon_c$) with increasing $r_0$. Derived SMF vs FMF slopes and intercepts confirmed

the general domination of $\varepsilon_c$ over $\varepsilon_f$ in controlling the "linear inverse" dynamic. The process of deriving and analyzing $\varepsilon_c$ and $\varepsilon_f$ values demonstrated an expected domination of FM optical depths for the urban pollution and biomass burning sites of GSFC and Mongu and thus a convergence towards the $\tau_f' = \tau_f$ ($\eta' = \eta$) line as $\tau_a$ increased (the convergence of SDA FM AODs towards AERinv FM AODs).

The more general conclusion resulting from this analysis is the apparent empirical confirmation that the influence of

PSD modal features can be detected by an indirect comparative analysis. While one would like to believe that this is true in general, a more comprehensive event-level closure experiment employing, for example, multi-altitude microphysical and optical measurements over a representative suite of AERONET instruments would do much to increase the level of confidence in such a conclusion.

## Appendix A

### A.1 Frequency of occurrence analyses

The retrievals results in this subsection are restricted to the two sites that are strongly impacted by historically large variations in FM particles (GSFC and Mongu). These sites (details in Table 1) can experience $\tau_f$ values greater than unity.

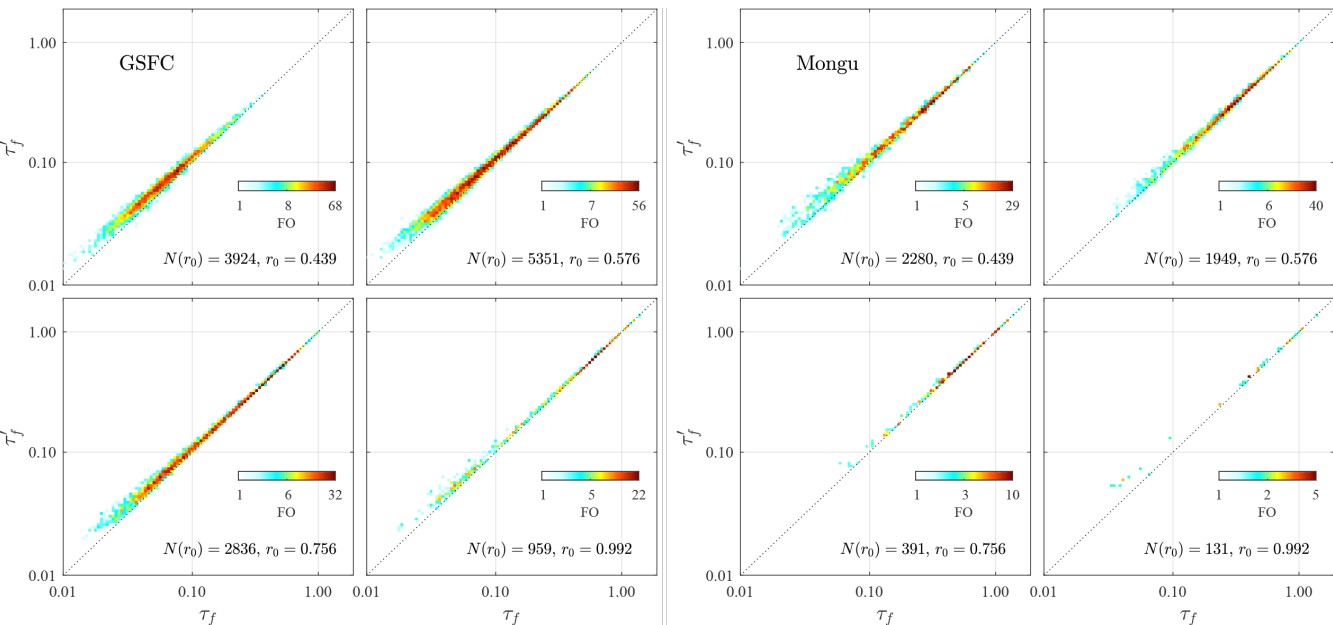

**Figure A1:** $\eta' - \eta$ vs $\tau_a$ FO distributions for GSFC and Mongu.





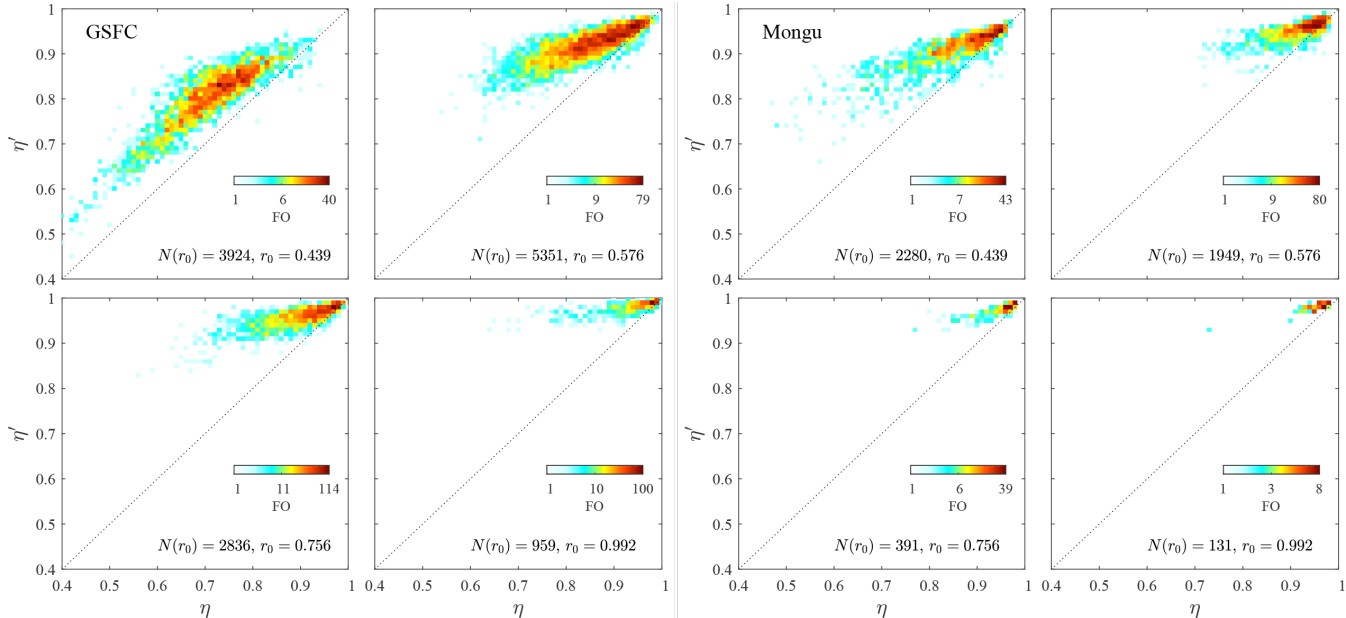

**Figure A2:** $\eta'$ vs $\eta$ FO distributions for GSFC and Mongu.

## A.2 Dual-modal optical equivalency of a tri-modal size distribution

### A.2.1 Fine mode and two coarse modes

One often sees a bi-modal coarse mode PSD and thus a tri-modal PSD. Treating the bi-modal CM PSD as an optically equivalent single mode CM PSD can be formalized as below (optical equivalent of equation (1a) of O'Neill et al., 2003):

$$\tau_a = \tau_f + \tau_c = \tau_f + \tau_{c1} + \tau_{c2} = \tau_f + \eta_{c1}\tau_c + \eta_{c2}\tau_c \qquad (A1)$$

where $\eta_{c1} = \frac{\tau_{c1}}{\tau_c}$, $\eta_{c2} = \frac{\tau_{c2}}{\tau_c}$. Accordingly;

$$\tau_a = \tau_f + \eta_{c1}\tau_c + (1 - \eta_{c1})\tau_c \qquad (A2)$$

Equations (3) of O'Neill et al. (2003) can be written as;

$$\alpha\tau_a = \alpha_f\tau_f + \alpha_{c1}\tau_{c1} + \alpha_{c2}\tau_{c2}$$

$$\alpha\tau_a = \alpha_f\tau_f + \tau_c(\alpha_{c1}\eta_{c1} + (1 - \eta_{c1})\alpha_{c2})$$

$$= \alpha_f\tau_f + \tau_c\langle\alpha_c\rangle \qquad (A3)$$

where $\langle\alpha_c\rangle = \alpha_{c1}\eta_{c1} + (1 - \eta_{c1})\alpha_{c2}$ represents the optical average of $\alpha_{c1}$ and $\alpha_{c2}$. Equation (A3) is exactly the analogue of O'Neill et al., 2003). Differentiating and recalling that $d\tau/d\ln\lambda = -\alpha\tau$, one finds;

$$\alpha'\tau_a = \alpha'_f\tau_f + \alpha'_{c1}\tau_{c1} + \alpha'_{c2}\tau_{c2} + \alpha^2\tau_a - \alpha_f^2\tau_f - \alpha_{c1}^2\tau_{c1} - \alpha_{c2}^2\tau_{c2}$$

$$= \alpha'_f\tau_f + (\alpha'_{c1}\tau_{c1} + \alpha'_{c2}\tau_{c2}) + \{\alpha(\alpha\tau_a)\} - \alpha_f^2\tau_f - (\alpha_{c1}^2\frac{\tau_{c1}}{\tau_c} - \alpha_{c2}^2\frac{\tau_{c2}}{\tau_c})\tau_c$$

$$= \alpha'_f\tau_f + \langle\alpha'_c\rangle\tau_c + \{\alpha\alpha_f\tau_f + \alpha\langle\alpha_c\rangle\tau_c\} - \alpha_f^2\tau_f - \langle\alpha_c^2\rangle\tau_c$$





$$= \alpha_f' \tau_f + \langle \alpha_c' \rangle \tau_c + \alpha_f \tau_f (\alpha - \alpha_f) + \langle \alpha_c \rangle \tau_c (\alpha - \langle \alpha_c \rangle) + \langle \alpha_c \rangle^2 \tau_c - \langle \alpha_c^2 \rangle \tau_c$$

but since $\alpha = \alpha_f \eta + (1 - \eta)\langle \alpha_c \rangle$ we can replace $(\alpha - \alpha_f)$ and $(\alpha - \langle \alpha_c \rangle)$ to obtain;

$$= \alpha_f' \tau_f + \langle \alpha_c' \rangle \tau_c - \alpha_f \tau_f (1 - \eta)(\alpha_f - \langle \alpha_c \rangle) + \langle \alpha_c \rangle \tau_c \eta(\alpha_f - <\alpha_c>) + \tau_c(\langle \alpha_c \rangle^2 - \langle \alpha_c^2 \rangle)$$

$$= \alpha_f' \tau_f + \langle \alpha_c' \rangle \tau_c - (1 - \eta)\tau_f (\alpha_f - \langle \alpha_c \rangle)^2 + \tau_c(\langle \alpha_c \rangle^2 - \langle \alpha_c^2 \rangle), \text{ so that;}$$

$$\alpha' = \eta \alpha_f' + (1 - \eta)\{\langle \alpha_c' \rangle + \langle \alpha_c \rangle^2 - \langle \alpha_c^2 \rangle\} - (1 - \eta)\eta(\alpha_f - \langle \alpha_c \rangle)^2$$

These expressions are optically equivalent to equation (5) of O'Neill et al. (2003) with their equation (5) values being transformed according to;

$$\alpha_c \leftrightarrow \langle \alpha_c \rangle$$

$$\alpha_c' \leftrightarrow \langle \alpha_c' \rangle + \langle \alpha_c \rangle^2 - \langle \alpha_c^2 \rangle$$

where the average value of any parameter $x_c$ is always;

$$\langle x_c \rangle = x_{c1} \eta_{c1} + x_{c2}(1 - \eta_{c1})$$

**A.2.2 Coarse mode and two fine modes**

In this instance one can imagine two fine mode components: for example (i) large-sized FM smoke and (ii) smaller sized FM urban-industrial pollution. The algebra above for the single FM PSD and two CM PSDs can be employed here: everything is perfectly reversible (interchange index c with index f) if η is viewed as the ratio of coarse to total AOD. Then arriving at the final equations for α and α' one can return to the usual definition of η to obtain the same algebraic theorem. The classic equations for $\alpha$ and $\alpha'$ remain unchanged providing one employs the substitutions below;

$$\alpha_f \leftrightarrow \langle \alpha_f \rangle$$

$$\alpha_f' \leftrightarrow \langle \alpha_f' \rangle + \langle \alpha_f \rangle^2 - \langle \alpha_f^2 \rangle$$

where $\langle x \rangle = \eta_f x + (1 - \eta_f)x$ and $\eta_f = \tau_{f,1}/\tau_f$. Explicitly;

$$\alpha = \alpha_c(1 - \eta) + \eta\langle \alpha_f \rangle$$

$$\alpha' = (1 - \eta)\alpha_c' + \eta\{\langle \alpha_f' \rangle + \langle \alpha_f \rangle^2 - \langle \alpha_f^2 \rangle\} - (1 - \eta)\eta(\langle \alpha_f \rangle - \alpha_c)^2$$

**Symbol and acronym glossary**

| | |
|---|---|
| AEROCAN | Federated Canadian subnetwork of AERONET run by Environment and Climate Change Canada (ECCC) |
| AERONET | Aerosol Robotic Network: World-wide NASA network of combined sunphotometer / sky-scanning radiometers manufactured by CIMEL Éléctronique.<br>See http://aeronet.gsfc.nasa.gov/ for documentation and data downloads |
| AOD | Aerosol optical depth: The community uses "AOD" to represent anything from nominal aerosol optical depth which hasn't been cloud-screened to the conceptual (theoretical) interpretation of aerosol optical depth. In this paper we use it in the latter sense and apply adjectives as required. |



| | |
|---|---|
| AERinv | AERONET Inversion |
| SDA | Spectral Deconvolution Algorithm |
| CM | Coarse mode |
| FM | Fine mode |
| FO | Frequency of Occurrence |
| PSD | particle size distribution (precisely, the volume particle size distribution) |
| x | x = a, f, or c (total, fine mode or coarse mode) |
| FMF, $\eta$ | Fine Mode Fraction (an output parameter computed from SDA products) |
| SMF, $\eta'$ | Sub-Micron Fraction (an output parameter computed from AERinv products) |
| $r_0$ | Cut off radius (AERinv ''inflection point'') |
| $\tau_x$ | Aerosol or FM or CM AOD at 500 nm (SDA) |
| $\tau'_x$ | Aerosol or FM or CM AOD at 500 nm (AERinv) |
| $\epsilon_x$ | Cut off portion of the fine or coarse mode optical depth (c.f. equation 7b) |

*Author contributions*

NTO led the conceptualization, methodology and formal analysis. All co-authors contributed to the revisions of the manuscript. TFE, JSR and DMG provided fundamental comments that resulted in substantial changes to the manuscript. KR, LI, DPR and JPC were instrumental in the strategical planning and operational processing of the data and the definition of the figures.

*Data availability*

All AERinv and SDA retrieval data are available from the AERONET website.

*Competing interests*

The authors declare that they have no conflict of interest.

*Acknowledgements*

We acknowledge the generous permission of the AERONET PIs[13] for the use of their AERinv and SDA retrieval data.

*Financial support*

This work was supported by CANDAC (the Canadian Network for the Detection of Atmospheric Change) through PAHA (Probing the Atmosphere of the High Arctic) funding (RGPCC-433842-2012), the NSERC Discovery Grant program (O'Neill, RGPIN-2017-05531), GSFC / NASA, notably GESTAR2 funding for O'Neill (SURA-GSTR-4100-NASA) and CSA's ESS-
DA program (16SUASACIA) and CSA's FAST program (CASSAVA – PEARL project led by K. Strong of CANDAC).

---

[13] Brent Holben of AERONET/NASA (Goddard Space Flight Center) at the Hamim, GSFC, Lanai, Midway Island, Solar Village, Thule and Mongu sites, Ihab Abboud and Vitali Fioletov of AEROCAN/Environment and Climate Change Canada (ECCC) at the PEARL site, Richard Wagener of Brookhaven National Laboratory (BNL) at the Barrow site and Andrew Clive Banks of Hellenic Centre for Marine Research (HCMR) for the Forth Crete site.



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
