# Peer review of "Relationship between the sub-micron fraction (SMF) and fine mode fraction (FMF) in the context of AERONET retrievals"

_Atmospheric Measurement Techniques, 2022_

## Author Response (AR1)

**Response to Referee Comment #1**

*The referee's comments are presented in italic* and our answers are written in plain text. **Modifications of the manuscript, if any, are written in plain bold text.**

*In this paper "Relationship between the sub-micron fraction (SMF) and fine mode fraction (FMF) in the context of AERONET retrievals", the authors explored differences between two AERONET products: sub-micron fraction from the AERONET inversion method and fine mode fraction (FMF) from the Spectral Deconvolution Algorithm. This is a well written paper with both the methods and results nicely presented. Also, FMF/SMF are widely used for discriminating fine from coarse mode aerosols, and thus this paper shall be of interest to AERONET data users as well as other readers. I recommend publication of this paper after some minor changes as listed below.*

We thank the referee for the insightful feedback.

*Comments:*

1. *AERONET data may be subject to thin cirrus cloud contamination. How would thin cirrus cloud contamination affect results as illustrated in this study?*

As mentioned in line 199 (page 9) of our initial submission, all the data used is Version 3 Level 2.0. This is a cloud screened product, as mentioned at:
https://aeronet.gsfc.nasa.gov/new_web/data_description_AOD_V2.html
The AERONET cloud screening criteria are explained at:
https://aeronet.gsfc.nasa.gov/new_web/Documents/Cloud_scr.pdf

If, despite the screening criteria, some data is contaminated with very thin cirrus cloud, the cirrus will simply contribute to the coarse mode optical interpretation (as a large coarse mode aerosol). The analysis that is carried out in this paper will not change in any essential way.

2. *This is more of a thought than a request for changes. Log-normal size distributions are assumed for fine mode and coarse mode aerosols in this study. However, aerosol particle size distributions may not be perfect log-normal. Could some of the observed differences between FMF and SMF be attributed to non-perfect log-normal distributions?*

The SDA-derived FMF retrieval is a purely spectral (optical) retrieval while the AERinv PSD is perfectly general (aside from the constraints of finite bin size): lognormal fits are not part of the overarching arguments of the paper. The log-normal distribution employed in Figures 1 and 2 are only illustrative. We see that the sentence we wrote (in the discussion of Figure 2) wasn't clear enough: so we tried to clarify it:

**We emphasize that lognormal fits to AERinv PSDs are not part of the empirical analysis process presented in this paper (nor do they have any role in the purely spectral SDA retrieval): rather the purpose of Figure 2 is to confirm the expected strong correlation between optical and**

**microphysical CM cutoff fractions and thereby facilitate an understanding of $\varepsilon_c$'s role in the dynamics of equation (7a).**

3. *Page 160, "Figure 2 is a plot of $\varepsilon_c$ vs $\Delta S_c/S_c$ for a variety of retrievals from the sites listed in Table 1". Could the authors be more specific about "a variety of retrievals"? Any criteria for selecting retrievals from the sites listed in Table 1? It is also confusing as the Figure 2 caption seems to indicate data points shown in Figure 2 are "simulated optical depth retrievals".*

Figure 2 is based on log-normal distributions and exact Mie calculations (for which we controlled all microphysical and optical parameters) that matched, as closely as possible, the actual AERinv and SDA inputs and retrievals. This allowed for exact (Mie- and lognormal-based) calculations of the $\varepsilon_c$ and $\Delta S_c/S_c$ values in Figure 2. The only selection criteria for all the cases of Figure 2 was to cover a large $\varepsilon_c$ range (a large variety of coarse mode aerosol types).

4. *Figure 3, "1 - $\varepsilon_f$" shall be "$1 - \varepsilon_c$ - $\varepsilon_f$" ?*

No, when $\eta$ = 1, at the extreme right of the Figure 3 scattergram, then $\eta' = (1 - \varepsilon_c - \varepsilon_f) \eta + \varepsilon_c = 1 - \varepsilon_c - \varepsilon_f + \varepsilon_c = 1 - \varepsilon_f$

5. *Line 200, "within a time window of ± 16 minutes about the nominal". Any reason for picking +- 16 minutes as the temporal window? Do the authors considered temporal homogeneity in spectral AODs in their data selection steps?*

The selected time window is a trade-off between insuring temporal homogeneity and maximizing the number of SDA retrievals (to attain statistically meaningful average whose natural variation is minimal). It was a time window recommended by AERONET co-author Tom Eck.

6. *Line 207, need a citation for the SDA method.*

The SDA was referenced in lines 46-47 of the submitted manuscript.

7. *Line 300, "The values corresponding to the four red-colored". It looks like 5 red-colored circles above the $\varepsilon_c$=0 line to me.*

Corrected in the new draft.

8. *Figure A1 caption. "$\eta' - \eta$ vs $\tau_f$" shall be $\tau_f$ vs $\tau_f'$ "?*

Yes, this was corrected in the new draft with the caption:

**$\tau_f'$ vs $\tau_f$ FO distributions for GSFC and Mongu. The FO colour scale is tied to variations on a logarithmic scale (with an attendant tendency to enhance the contributions of large FO values). N($r_0$) is the total FO at a given $r_0$ value.**

**Response to Referee Comment #2**

*The referee's comments are presented in italic* and our answers are written in plain text. **Modifications of the manuscript, if any, are written in plain bold text.**

*The authors presented a simple sub-micron fraction versus fine mode fraction linear equation that makes it possible to better understand the well recognized empirical result of SMF being greater than FMF. The paper is well-constructed, and the statistical method is serious analyzed and has scientific value. Overall, publication is recommended after addressing the following minor revisions.*

We thank the referee for the insightful feedback.

*Comments:*

1. *Line 91: "governed by that relationship", Maybe it would be better for the reader to understand the paper by stating in the text which relationship is governed by.*

   We changed the sentence to the version below in an effort to clarify our meaning:
   **We seek to demonstrate that the AERinv-derived value of SMF and the SDA-derived value of FMF are largely linked by that simple relationship and that fitting parameters extracted from their empirical comparison yields insight into their fundamental opto-physical dynamics.**

2. *Table1 and line 208-212: for the classification of aerosol types at AERONET sites, maybe some citations for aerosol types needed here as a basis for the classification of different aerosol types.*

   A citation column was added to Table 1.

Norm, do you know any publication emphasizing the type of aerosols, at least for some of those sites?

3. *Figure 3: Maybe it should be "$1 - \varepsilon_c - \varepsilon_f$" instead of "$1 - \varepsilon_f$"?*

No, when $\eta = 1$, at the extreme right of the Figure 3 scattergram, then $\eta' = (1 - \varepsilon_c - \varepsilon_f)\, \eta + \varepsilon_c = 1 - \varepsilon_c - \varepsilon_f + \varepsilon_c = 1 - \varepsilon_f$

4. *Line 213: First line indent.* √
5. *Line 244: First line indent.* √
6. *Line 253, 261 and 273: figure S1 was not found in the paper.*

Figure S1 is part of the "supplementary material" submitted with this paper.
To make sure this is clear, at the first Figure S1 instance, the sentence

(scattergrams for the rest of the aerosol types and sites can be seen in Figure S1)
became in the new draft
(scattergrams for the rest of the aerosol types and sites can be seen in Figure S1 **of the supplementary material**)

7. *Line 275: As well, Figure S2 was not found in the paper*

Figure S2 is part of the "supplementary material" submitted with this paper. The Figure S1 clarification above applies equally well to Figure S2.

8. *Figure A1 caption: maybe it should be "$\tau_f$ vs $\tau_f'$" instead of "$\eta' - \eta$ vs $\eth$⬚⬚⬚$_{\eth}$⬚⬚⬚"?*

Yes, this was corrected in the new draft with the caption:

$\tau_f'$ **vs** $\tau_f$ **FO distributions for GSFC and Mongu. The FO colour scale is tied to variations on a logarithmic scale (with an attendant tendency to enhance the contributions of large FO values). N($r_0$) is the total FO at a given $r_0$ value.**

---

## Author Response (AR2)

**Response to Referee Comment #3**

*The referee's comments are presented in italic* and our answers are written in plain text. **Modifications of the manuscript, if any, are written in plain bold text.**
*The manuscript by O'Neill et al. present a simple linear equation between SMF and FMF which explains why SMF is generally greater than the FMF. The manuscript is thorough, clear and well written. I only have some minor/technical comments as listed below.*

We thank the referee helpful feedback.

*Comments:*

1. *line 70: "Kaku et al., 2014 (Ka)" mentioned only here and not included in the references.*

We added the reference.

2. *line 172 "SMF being generally >~ ≳ FMF" seems to have an issue with a symbol or character, at least on my system.*

We removed "≳". This was an accidental insertion.

3. *line 200: Could the average be calculated from a single AOD observation or was there a requirement for a minimum number of observations?*

A footnote was added to the sentence defining the "time window of $\pm$ 16 minutes":
"**The number of AODs allowed to define a match in the time window could be as small as one (i.e. there was no minimum number).**"

4. *line 209: Why did you choose specifically these sites and not other sites with similar aerosol types (e.g. Alta Floresta)? Biomassa burning events are limited to specific seasons, so did you only consider the burning season in Mongu or did you use all the available data?*

A relevant point. A sentence was added to the end of the small paragraph describing our choice of aerosol classes:

"**Our choice of aerosol types was not intended to be comprehensive from the standpoint of investigating variations in say, different types of smoke aerosols or different types of dust aerosols: rather we sought to properly exercise the SMF versus FMF relationship by largely filling the admissible portion of the SMF versus FMF scattergram (see Figure 3).**"

5. *line 220: "M" seems to be mentioned here for the first time but it has not been defined anywhere.*
"(large M)" was changed to "**(large solar airmass, M)**"

6. *Figure 6: Please, clarify in the caption the meaning of the color in the points.*

Somehow we failed to fix that obvious oversight. There is now a new 2^nd sentence in the Figure 6 caption:

"**The four colors represent the four different AERinv inflection points ($r_0$ values with units of μm).**"

7. *Figure 9: "The orange filled large circle represents the only $\varepsilon f$ point... " I couldn't find any orange circle from the figures. Could you make it more apparent?*

The figure was incorrect : the present figure includes an "**orange-filled (red-rimmed) large circle**".

8. *Figure A2: I would prefer if the full caption of the figure would be presented here so that the reader does not have to go to the supplement to find it.*

That was a mistake. The Figure A2 caption now refers to the caption of Figure A1 for details.

[revised manuscript text omitted]
525 the Aerosol Robotic Network (AERONET).)., Journal of Geophysical Research: Atmospheres, 117(D07206), https://doi.org/10.1029/2011JD016839, 2012.

  Eck TF,T.F., Holben BN,B.N., Giles DM,D.M., Slutsker I,., Sinyuk A,., Schafer JS,J.S., Smirnov A,., Sorokin M,., Reid JS,J.S., Sayer AM,A.M., Hsu NC,N.C., AERONET remotely sensed measurements and retrievals of biomass burning aerosol optical properties during the 2015 Indonesian burning season. Journal of Geophysical Research: Atmospheres,
530 124(8):4722-40, https://doi.org/10.1029/2018JD030182, 2019.

  Eck TF,T.F., Holben BN,B.N., Kim J,., Beyersdorf AJ,A.J., Choi M,., Lee S,., Koo JH,J.H., Giles DM,D.M., Schafer JS,J.S., Sinyuk A,., Peterson DA,D.A., Influence of cloud, fog, and high relative humidity during pollution transport events in South Korea: Aerosol properties and PM2.5 variability. Atmospheric Environment, 232:117530, https://doi.org/10.1016/j.atmosenv.2020.117530, 2020.

[revised manuscript text omitted]